# Dopamine neuron ensembles signal the content of sensory prediction errors

Thomas A Stalnaker[1][†]*, James D Howard[2][†], Yuji K Takahashi[1],
Samuel J Gershman[3], Thorsten Kahnt[2,4,5][‡], Geoffrey Schoenbaum[1,6,7][‡]*

[1]Intramural Research Program, National Institute on Drug Abuse, National Institutes of Health, Baltimore, United States; [2]Department of Neurology, Feinberg School of Medicine, Northwestern University, Chicago, United States; [3]Department of Psychology and Center for Brain Science, Harvard University, Cambridge, United States; [4]Department of Psychiatry and Behavioral Sciences, Feinberg School of Medicine, Northwestern University, Chicago, United States; [5]Department of Psychology, Weinberg College of Arts and Sciences, Northwestern University, Chicago, United States; [6]Department of Anatomy and Neurobiology, University of Maryland School of Medicine, Baltimore, United States; [7]Department of Neuroscience, Johns Hopkins School of Medicine, Baltimore, United States

*For correspondence:
thomas.stalnaker@nih.gov (TAS);
geoffrey.schoenbaum@nih.gov (GS)

[†]These authors contributed equally to this work
[‡]These authors also contributed equally to this work

**Abstract** Dopamine neurons respond to errors in predicting value-neutral sensory information. These data, combined with causal evidence that dopamine transients support sensory-based associative learning, suggest that the dopamine system signals a multidimensional prediction error. Yet such complexity is not evident in the activity of individual neurons or population averages. How then do downstream areas know what to learn in response to these signals? One possibility is that information about content is contained in the pattern of firing across many dopamine neurons. Consistent with this, here we show that the pattern of firing across a small group of dopamine neurons recorded in rats signals the identity of a mis-predicted sensory event. Further, this same information is reflected in the BOLD response elicited by sensory prediction errors in human midbrain. These data provide evidence that ensembles of dopamine neurons provide highly specific teaching signals, opening new possibilities for how this system might contribute to learning.
DOI: https://doi.org/10.7554/eLife.49315.001

## Introduction

Midbrain dopamine neurons are widely proposed to signal value prediction errors (*Mirenowicz and Schultz, 1994*). However, the same neurons also respond to errors in predicting the features of rewarding events, even when their value remains unchanged (*Howard and Kahnt, 2018*; *Takahashi et al., 2017*). Such sensory prediction errors would be useful for learning detailed information about the relationships between real-world events (*Gardner et al., 2018*; *Howard and Kahnt, 2018*; *Langdon et al., 2018*; *Takahashi et al., 2017*). Indeed, dopamine transients facilitate learning such relationships, independent of value, when they are appropriately positioned to mimic endogenous errors (*Chang et al., 2017*; *Keiflin et al., 2019*; *Sharpe et al., 2017*). Yet dopaminergic responses to sensory prediction errors do not seem to encode the content of the mis-predicted event, either at the level of individual neurons or summed across populations (*Howard and Kahnt, 2018*; *Takahashi et al., 2017*).

How then do downstream areas that receive this teaching signal know what to learn? The conventional response is that such signals are permissive, with downstream areas controlling the content of the resultant learning (*Glimcher, 2011*). However, another possibility is that information about the

content of the learning might be contained, at least partly, in the pattern of firing across ensembles of dopamine neurons. It is now widely accepted that information is represented in areas like cortex and hippocampus not by individual neurons, but rather in a distributed fashion in the firing of groups of cells (*Gochin et al., 1994*; *Jennings et al., 2019*; *Jones et al., 2007*; *Rich and Wallis, 2016*; *Rigotti et al., 2013*; *Schoenbaum and Eichenbaum, 1995*; *Wikenheiser and Redish, 2015*; *Wilson and McNaughton, 1993*). If this is true for the cortex and hippocampus, then why not for the midbrain dopamine system? Consistent with this, here we show that the pattern of firing across a small group of dopamine neurons recorded in rats contains specific information about the identity of a mis-predicted event. We further show that this same content-rich signal is evident in the BOLD response elicited by sensory prediction errors in human midbrain. These data provide the first evidence of which we are aware that dopamine neuron ensembles generate firing patterns capable of conveying not only the occurrence of a prediction error to downstream areas but also information regarding what exactly was mis-predicted. These findings open new possibilities for how dopaminergic error signals might contribute to the learning of complex associative information.

## Results

To address whether dopamine neurons function as an ensemble to represent sensory prediction errors, we analyzed data from rats trained on a variant of the odor-guided choice task used to demonstrate the joint signaling of value and sensory prediction errors in our prior report (*Takahashi et al., 2017*) (while a limited analysis of a subset of these data were presented in a supplemental section of our prior report, this is the first presentation of the full dataset and its analysis as an ensemble). In the task variant (*Figure 1a*), two fluid wells delivered either one or three drops of discriminable but equally-preferred solutions of grape or tropical punch Kool Aid. Rats initiated each trial with a nose-poke into an odor port. After a brief delay, one of two odors was presented, indicating that reward would be available in the left or right well on that trial. If the rat responded at the proper fluid well, the reward was delivered. To induce prediction errors to correlate with neural activity, reward number or flavor were manipulated across a series of four transitions between five trial blocks in each recording session. At the first and second transitions, rewards were omitted and delivered unexpectedly, respectively, to allow identification of classic reward prediction

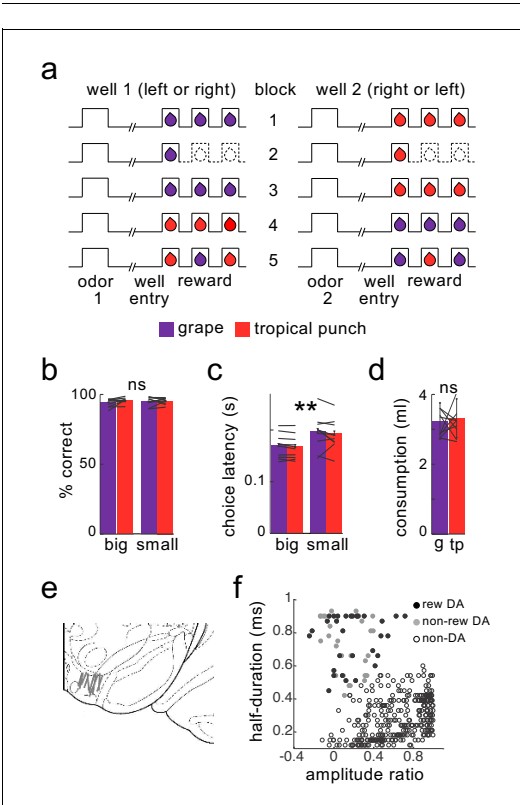

**Figure 1.** Task design and behavior during recording. Schematic (**a**) illustrates the order of events in trials at each well and the number and type of reward delivered at each well in the five trial-blocks performed in all recording sessions. Dashed lines indicate the omission of drops previously delivered. Rats were highly accurate in choosing the rewarded well during recording (**b**), and accuracy was unaffected by the flavor or number of drops at a particular well, either for the group or for individual subjects (flavor: $F_{1,193}$=1.3, p=0.26; number: $F_{1,193}$=1.0, p=0.32; interactions with subject: F's <= 1.0, p's > 0.47). Rats were faster to respond for the 3-drop rewards (**c**), and this effect was again unaffected by the flavor of reward, either for the group or for individual subjects (main effect of number: $F_{1,193}$=190, p<10$^{-6}$; main effect of flavor: $F_{1,193}$=1.75, p=0.19; flavor X subject interaction: $F_{9,193}$=0.86, p=0.56). A two-bottle preference test run at the end of the sessions (**d**) also revealed no effect of flavor ($F_{1,9}$=0.17, p=0.69). Data for individual subjects is illustrated by lines; error bars represent standard errors across sessions for percent correct and latency and across rats for the consumption test. Recordings were made in ventral tegmental area (**e**), and dopaminergic neurons (n = 30) were identified by waveform cluster analysis (**f**). **p<0.01. g = grape, tp = tropical punch.
DOI: https://doi.org/10.7554/eLife.49315.002

errors. At the third and fourth transitions, reward number remained constant, but flavor was changed. At one transition, the flavor of all three drops were changed to replicate what was done previously, while at the other, only one drop of the three changed, leaving the others unchanged to provide a control condition to distinguish signaling of flavor errors from signaling of flavor itself.

Neural activity in VTA was recorded using drivable bundles of microelectrodes. During recording, the rats were highly accurate, responding correctly on ~95% of the forced-choice trials, indicating that they had learned the meaning of the odor cues, independent of reward number or flavor (*Figure 1b*). The rats also exhibited an appreciation of the reward number, responding significantly faster when the 3-drop reward was at stake, an effect that was also independent of the reward flavor (*Figure 1c*). Indeed, choice latency was similar across the two flavors, even in the behavior of individual rats, suggesting that they valued the two flavors similarly in the task (*Figure 1c*, lines). This is consistent with preference testing conducted separately after recording, which indicated that individually and as a group the rats had no significant preference between the two flavors of Kool-Aid (*Figure 1d*).

Using waveform characteristics and firing in response to reward, as in previous papers (see Materials and methods), we identified 30 putative dopaminergic neurons recorded during these sessions (*Figure 1e and f* and *Table 1*). As previously reported (*Takahashi et al., 2017* in Supplemental Figure 2), the firing of these neurons exhibited classic reward prediction error correlates, decreasing in response to reward omission at the first transition and increasing in response to unexpected reward at the second transition, changes that were inversely correlated across neurons (*Figure 2a–c*). This is as expected based on numerous prior reports that individual dopamine neurons signal bidirectional errors in the prediction of reward, in different species, tasks, and labs (*Schultz, 2016*).

In addition, however, the same neurons also responded with elevated firing across transitions in which there was a change in reward flavor, combining both the third transition, presented previously (*Takahashi et al., 2017* in Supplemental Figure 2), and the more selective fourth transition, included here. These changes in firing occurred even though the rats' behavior – both in the task and in separate preference testing (*Figure 1b–d*) – indicated no difference in the subjective value of the two flavors, even for individual subjects. The dopamine neurons increased firing to changes in flavor, and the size of these increases were positively correlated between the two flavor errors (*Figure 2d and e*). Further, individual neurons showed very little difference between initial firing rates in response to the two different flavor errors (*Figure 2f*). Thus, the activity of these neurons, individually or on average, signaled that something unexpected had happened, but it did not distinguish details of that event (e.g. if grape was switched for tropical punch or vice versa).

To test whether such information might be available in the pattern of firing across a group or ensemble of dopamine neurons, we aligned activity from all neurons on like trials from each block, and then used a 'training set' of trials from each flavor-switch block to identify the ensemble pattern characteristic of the neural response to each flavor change. Individual trials left out of this training set were then matched to the two patterns in an attempt to decode the flavor that had been delivered. To assess the evolution of information coding within and across trials, we used a sliding time window aligned to events in a trial and a sliding window of trials that progressed across each block. The results indicated that the pattern of activity across the ensemble did contain information about

**Table 1.** Numbers of putative dopamine neurons recorded in each subject (subjects without dopamine neurons are not listed).

| Rat ID | # Dopamine Neurons |
| --- | --- |
| AA01 | 6 |
| AA05 | 9 |
| AA06 | 1 |
| AA07 | 4 |
| AA09 | 3 |
| AA10 | 1 |
| AA12 | 6 |

DOI: https://doi.org/10.7554/eLife.49315.004

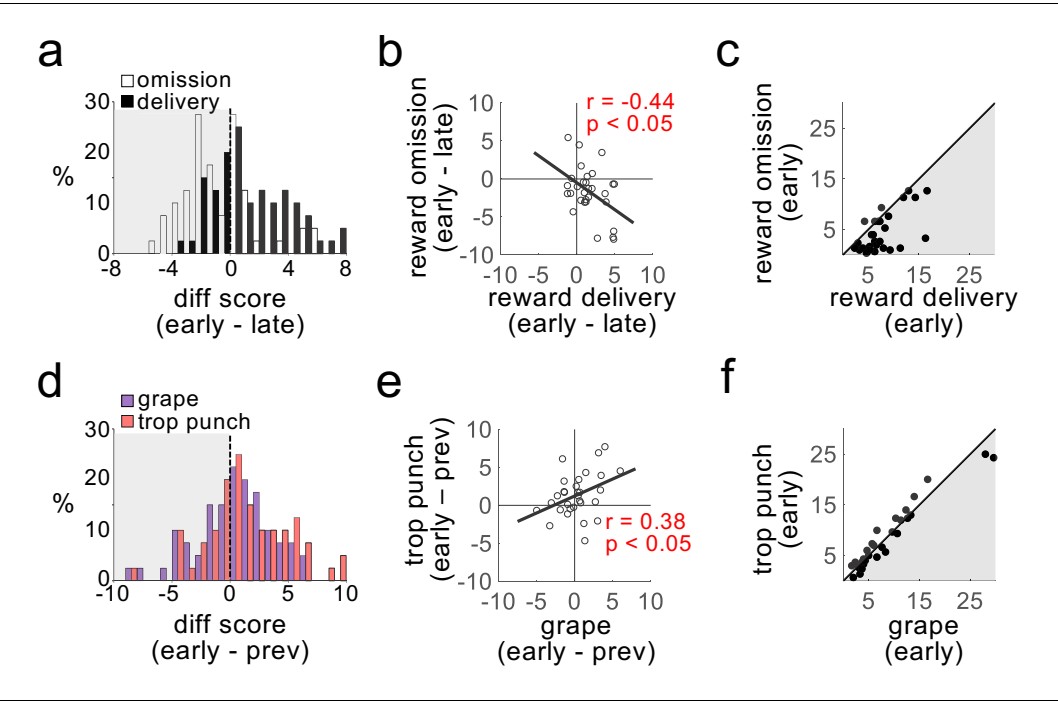

**Figure 2.** Dopamine neurons do not distinguish the identity of sensory prediction errors. Plots show firing rates of dopamine neurons in response to transitions in number of reward drops (omission or delivery; **a–c**) and flavor (grape or tropical punch; **d–f**). Changes in firing rate in response to omission (negative errors) and delivery (positive errors) were readily distinguishable (a; $t_{29}$ = 4.0, p<$10^{-3}$), inversely correlated across neurons (**b**), and firing rates were markedly different after the transition (c; $t_{29}$ = 5.2, p<$10^{-4}$). The same neurons exhibited increased firing rates in response to transitions in the expected flavor of reward (d); $t_{29}$ = 2.1, p<0.05), but the increases to the two flavors were indistinguishable ($t_{29}$ = −1.95, ns), positively correlated across neurons (**e**), and firing rates after the transition also did not distinguish the two flavor errors (f; $t_{29}$ = 0.13, ns).
DOI: https://doi.org/10.7554/eLife.49315.003

flavor in both of the flavor-change trial blocks (*Figure 3a and b*). Critically, however, accurate decoding of flavor was observed only for the drops where flavor had changed and then only on trials early in the blocks; accuracy was only seen in epochs immediately after the new drop was delivered and fell to chance later in the block, consistent with representation of the error in predicting the flavor – either the omission of the expected flavor or the delivery of the new flavor - and not representation of flavor itself.

This impression was confirmed when we formally compared decoding accuracy in time windows surrounding drops where the flavor had changed versus windows surrounding drops where the flavor had not changed. Accurate decoding was only observed when the drop had changed flavor, and then only in the first 10 trials of these blocks; decoding was best in the earliest trials immediately after the transition, fell to chance in the last 10 trials, and flavors from the early trials did not misclassify with the same flavors in the later trials (*Figure 3c and d*). Separate analyses indicated that flavor could be decoded from neural activity in these early trials as early as 175 ms after fluid delivery (see Materials and methods for details of analysis). The decline in decoding accuracy across the block occurred without any gross changes in baseline firing rates (*Figure 3d*). Thus, the dopamine neuron ensemble was representing not the flavor itself, but flavor when it had been mis-predicted.

Finally, as an additional test of this idea, we applied a similar approach to examine encoding of the information content of sensory prediction error signals previously reported in fMRI data in the human midbrain (*Howard and Kahnt, 2018*) (while these data were analyzed for sensory errors in our prior report, this is the first presentation of an MVPA analysis of these data to attempt to distinguish the content of the error signal). These data were collected from subjects performing a task in which they learned that abstract visual cues predicted the odors of different sweet (SW) and savory

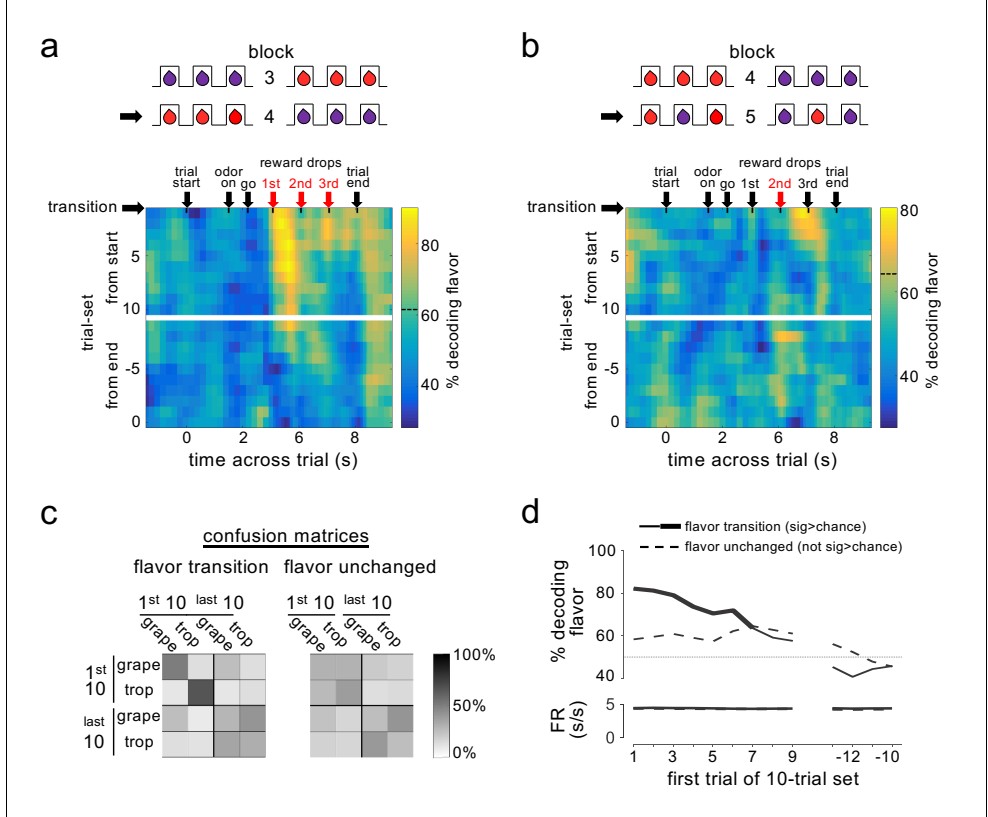

**Figure 3.** Dopamine ensembles distinguish the identity of sensory prediction errors. Heat plots show decoding of flavor by dopamine neuron ensembles, using data from a sliding window during trials after all three drops changed flavor (a) or when only the second drop changed flavor (b). Red arrows indicate the time of the new flavor drop delivery. In each case, decoding was significantly above chance at the changed drops, but only early in the block (dotted lines on scale bars show one-tailed 95% confidence interval upper bounds for chance, by permutation tests). This effect was also evident when we collapsed data from the two blocks and compared decoding in epochs capturing firing to the drops where flavor changed versus control epochs capturing firing where flavors had not changed (c); flavor could be decoded accurately by dopamine ensembles only immediately after changes in flavor (patterns in confusion matrices were significantly different at $p < 10^{-4}$ by permutation test). A more detailed analysis using sliding sets of 10-trials (d) showed the decay of flavor decoding as the block progressed (upper plot, solid line), while control decoding of flavor (dotted line) and baseline firing rates in both conditions (lower plot) were unchanged across the block. Thick line in the upper plot shows significance compared to chance ($p < 0.05$ for at least five significant trial sets by permutation test). Thin dotted line in upper plot shows chance decoding level.

DOI: https://doi.org/10.7554/eLife.49315.005

(SV) food odor rewards (*Figure 4a*). The rewarding odors were matched in value, as reflected in both pleasantness ratings acquired before the learning task (*Figure 4b*) and choices made during the task (*Figure 4c*). During the fMRI scanning session, the odors associated with the visual cues were switched across blocks of trials (i.e., SW→SV and SV→SW), thereby inducing value-neutral sensory prediction errors similar to those induced by the flavor switches in the rat task described above. Previously it was reported that these switches evoked prediction error-like responses in the BOLD signal in the midbrain (*Howard and Kahnt, 2018*; *Suarez et al., 2019*). Here we utilized a multivoxel pattern analysis (MVPA) to test whether distributed fMRI activity patterns in this region contained information about the content of the error immediately after a switch and then later after learning.

This analysis, which is conceptually similar to that applied to the single unit activity described above, found that it was possible to decode the identity (SW or SV) of the unexpected odor from the midbrain activity at the time the error was experienced (*Figure 4d*). Importantly, decoding was significantly above chance only on the trials in which the food odors were mis-predicted but at

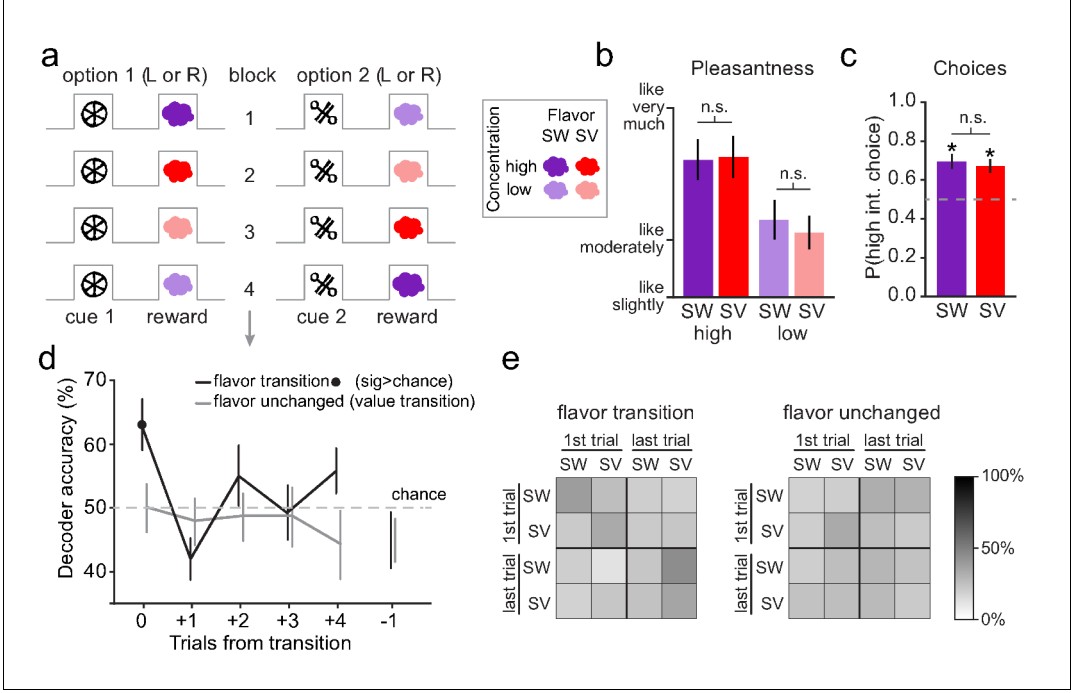

**Figure 4.** Human midbrain distinguishes the identity of sensory prediction errors. (a) The reversal learning task involved binary choices between two visual cues to receive either a high or low concentration of one of two food odor rewards (one sweet [SW] and one savory [SV]). The associations were covertly changed throughout the task to induce either sensory prediction errors (e.g. transition from block 1 to block 2) or value prediction errors (e.g. transition from block 2 to block 3). (b) Sweet and savory food odors were matched for pleasantness within each odor concentration (SW high vs. SV high: $t_{22}$ = 0.18, p=0.86; SW low vs. SV low: $t_{22}$ = 1.16, p=0.26). Error bars depict within-subject s.e.m. (c) On free choice trials, the cue associated with the high-concentration odor was chosen significantly above chance (50%) for both odor identities (SW: $t_{22}$ = 4.03, p=$2.83 \times 10^{-4}$; SV: $t_{22}$ = 4.20, p=$1.83 \times 10^{-4}$) and these choice proportions did not differ significantly from each other ($t_{22}$ = 0.71, p=0.48). Error bars depict within-subject s.e.m. (d) Decoding accuracy of SW vs. SV was significantly above chance on the error trial of flavor transitions (black line) ($t_{22}$ = 3.22, p=0.004), but not for subsequent trials or the trial preceding error trials (p's > 0.12). Decoding accuracy of SW vs. SV was at chance for the error trial on value transitions (gray line), as well as subsequent trials, and the trial preceding the value transitions (p's > 0.15). Error bars depict within-subject s.e.m. (e) Confusion matrices show the decoding accuracy for individual conditions within the decoding analyses (there was a trend that patterns in confusion matrices were different at p=0.08 by permutation test). Within the top left quadrant of the flavor transition matrix (i.e. training and testing the classifier on the error trial of flavor transitions), across all subjects and iterations, accuracy was at 63.3% for SW predictions and 63.8% for SV predictions. All other comparisons for flavor transitions and all comparisons for value transitions were at chance.
DOI: https://doi.org/10.7554/eLife.49315.006

chance on subsequent trials when food odors were delivered as expected (*Figure 4d*). Follow-up examination of the decoder performance confirmed that decoding was only above chance on the error trial, and that the decoder was not biased towards prediction of a particular odor (*Figure 4e*), consistent with representation of the mis-predicted food odors and not the food odors themselves.

## Discussion

The results presented here show that, in both rats and humans, putative dopaminergic sensory prediction error responses in the midbrain contain specific information about the features of the mis-predicted event itself, appropriate for instructing or updating representations in downstream brain regions. These results are consistent with the proposal that the midbrain dopamine system signals a multidimensional prediction error, able to reflect a failure to predict information about an unexpected event beyond and even orthogonal to value (*Gardner et al., 2018*; *Howard and Kahnt, 2018*; *Langdon et al., 2018*; *Takahashi et al., 2017*). Importantly this proposal is not necessarily

contrary to current canon; it can account for value errors as a special example of a more general function (*Gardner et al., 2018*), one readily apparent in the firing of individual neurons perhaps due to the priority given to such information when it is the goal of the experimental subject. However, this proposal also explains in a relatively straightforward way why dopamine neurons are often phasically active in settings where value errors were not anticipated a priori, at least by the experimenters, such as when novel cues or even information is first presented (*Bromberg-Martin and Hikosaka, 2009*; *Horvitz, 2000*; *Horvitz et al., 1997*; *Kakade and Dayan, 2002*), or even in response to violations in beliefs or auditory expectations (*Gläscher et al., 2010*; *Gold et al., 2019*; *Iglesias et al., 2013*; *Schwartenbeck et al., 2016*). That the pattern of firing across a relatively small population of dopamine neurons can provide details regarding the mis-predicted event endows the dopamine system with the ability to serve as an instructive 'teaching' signal outside the dimension of value.

One interesting question raised by the prior and current results is whether and how such a system would distinguish the omission of an expected sensory event from its unexpected appearance. The designs of the two experiments analyzed here do not allow us to distinguish representation of these two types of errors. We would speculate that both should be encoded in the neural activity of the system, including in the current data. Thus, the decoding demonstrated here would reflect the combination of these two changes. Of course, the actual presence of something is likely to support a much stronger signal than its absence, so in practice, it may be difficult or require substantially higher statistical power to see a representation of an omitted event, particularly one that involves subtle features orthogonal to value.

Another interesting question raised by these results is whether downstream areas use the information in the signal to support learning. While the current data is only correlative, it is notable that the information is only there when it is relevant to learning at the start of the blocks, so it is appropriately positioned to be of use to drive learning in downstream structures. And of course, a causal role for the signal shown here is in line with recent demonstrations that dopamine transients are necessary and sufficient for learning that cannot be easily accounted for by classic reinforcement learning mechanisms (*Chang et al., 2017*; *Keiflin et al., 2019*; *Sharpe et al., 2017*). *Keiflin et al. (2019)* is particularly relevant in this regard, since in this study, conditioned responding to a cue unblocked by artificial activation of VTA dopamine neurons at the time of an expected reward was shown to be sensitive to subsequent devaluation of that reward. Sensitivity to devaluation indicates that the artificial dopamine transients induced the formation of an association between the conditioned stimulus and the sensory properties (i.e. the flavor) of the reward, precisely the type of learning the signal here would be proposed to support (*Gardner et al., 2018*).

How the artificial activation of neurons engaged in representing information through a pattern of activity can cause normal learning in studies such as those cited above is another outstanding question raised by the current data. One possible explanation for this may be found in the appearance of external events at the time of stimulation in these studies. Even though these events are largely expected in the blocking designs used in *Sharpe et al. (2017)*; *Keiflin et al. (2019)*, input reflecting their appearance still impinges on the dopamine neuron population at the proper time to support learning. By randomly injecting current across a subset of this population, the artificial stimulation may recover a ghost of the error pattern that would be caused by these events if they were unexpected – a pattern close enough to cause learning that seems normal, given the very simple behavioral readouts used in these studies.

If dopamine neurons do provide information about errors beyond the single dimension of value, this brings up questions about the limits on this and how this system deals with the vastness of the possible error space relative to the number of dopamine neurons. There are approximately 40,000 dopamine neurons in the VTA of rats, and another 25,000 in SN (*Nair-Roberts et al., 2008*). In humans, the total number is about 300,000 (*Hirsch et al., 1988*). If each neuron provides only a single bit of information, the capacity of just the VTA in rats is still 240,000. Of course, there is surely substantial redundancy across neurons, yet even if we reduce the cell number to 1000 real bits of information, we still end up with 1.0715e+301 potential patterns. This is a huge number. And of course, information represented in spiking may be augmented (or attenuated) by factors such as co-release of other neurotransmitters downstream and the location (region, cell type, dendritic compartments) and type (receptors, second messenger cascades, interactions capable of modulating) of interactions with downstream regions, etc. Even if all this combines to yield only 20 or 30 unique coded dimensions, we still end up with a billion possible patterns of output. This number seems big

enough, with assistance from other systems (we do not propose this to be the only learning signal) and with contextual modulation of the processing (i.e. some factors might be given priority or not, depending on situation, by modulating inputs), to deal with much of the problem of dimensionality.

Finally it is worth noting that the demonstration here mirrors advances in the computational field, where distributed, multidimensional error signaling is a key component of more advanced algorithms, such as distributed reinforcement learning and successor representation (*Dabney et al., 2017*; *Dayan, 1993*). In both, the error driving learning is not unitary but rather is represented as a vector. Distributed reinforcement learning has recently been suggested as an explanation for the heterogeneity of the responses of individual dopamine neurons to errors in predicting reward value (*Kurth-Nelson et al., 2019*). The current results extend this to show for the first time that an assembly of dopamine neurons can function to represent the content of errors, even outside the realm of value. That the same information available in the pattern of activity is not readily apparent in the activity of individual neurons is in accord with ideas guiding behavioral neurophysiology in other areas (*Yuste, 2015*), and suggests it is time to consider the functions of the dopamine system across rather than within individual neurons.

# Materials and methods

## Experiment 1

### Subjects

Ten male Long-Evans rats (Charles River Labs, Wilmington, MA), aged approximately 3 months at the start of the experiment and single-housed once the experiment began, were used in this study. Rats were tested at the NIDA-IRP in accordance with NIH guidelines determined by the Animal Care and Use Committee.

### Surgical procedures

All surgical procedures adhered to guidelines for aseptic technique. For electrode implantation, a drivable bundle of eight 25 um diameter NiCr/Formvar wires (A-M Systems, Sequim, WA) chronically implanted dorsal to VTA in the left or right hemisphere at 5.2 mm posterior to bregma, 0.7 mm laterally, and 7.5 mm ventral to the brain surface at an angle of 5° toward the midline from vertical. Wires were cut with surgical scissors to extend ~2.0 mm beyond the cannula and electroplated with platinum ($H_2PtCl_6$, Aldrich, Milwaukee, WI) to an impedance of 800–1000 kOhms. Cephalexin (15 mg/kg p.o.) was administered twice daily for two weeks post-operatively.

### Histology

All rats were perfused with phosphate-buffered saline (PBS) followed by 4% paraformaldehyde (Santa Cruz Biotechnology Inc, CA). Brains were cut in 40 μm sections and stained with thionin and then examined to determine electrode placement.

### Behavioral task

Training and recording was conducted in aluminum chambers approximately 18' on each side with sloping walls narrowing to an area of 12' x 12' at the bottom. A central odor port consisting of a small hemicylinder accessible by nose-poke was located about 2 cm above two fluid wells, and higher up on the same wall were mounted two lights. The odor port was connected to an airflow dilution olfactometer to allow the rapid delivery of olfactory cues, which were chosen from compounds obtained from International Flavors and Fragrances (New York, NY). Trial availability was signaled by illumination of the panel lights inside the box. When these lights were on, a nosepoke into the odor port resulted in delivery of the odor cue for 500 ms. One of two different odors was delivered to the port on each trial in a pseudorandom order such that in each 50 trials there were 25 of each, and the same odor was never presented for more than three consecutive trials. At odor offset, the rat had 3 s to make a response at one of the two fluid wells. One odor indicated that reward would be available at the left well, while the other indicated it would be available at the right well; errors resulted in no reward delivery and the lights turning off (errors occurred on about 5% of trials across all recording sessions; see *Figure 1b*). On correct trials, lights turned off once rats had

finished licking at the well; the intertrial interval was ~2–3 s before the light turned on once again. Once the rats were shaped to respond accurately (at least ~75%) on both odors, we introduced trial-blocks in which the number and flavor of reward drops (one or three drops of Grape or Tropical Punch Kool-Aid solution) were constant within a block but changed between blocks according to the schedule summarized in *Figure 1a*. The drop volume was ~0.05 ml and multiple drops were delivered 1000 ms apart. For each recording session, wells were randomly designated such that in the first trial-block, correct responses at one well resulted in delivery of 3 drops of grape solution while correct responses at the other well resulted in 3 drops of tropical punch solution. In the second trial-block, the number of drops available on both sides changed from three to one, with the flavor remaining the same. In the third trial-block, the number of drops available on both sides changed from one back to three, again with the flavor remaining the same. On the fourth trial-block, the flavor of all three drops on each side were switched to the other flavor. Finally, in the fifth trial-block, the flavor of the second drop on each side was switched to the opposite flavor, with the other two on both sides remaining the same. Thus, in each session, there was one number downshift transition (drop omission), one number upshift transition (new drop deliveries), one flavor transition across all three drops, and one flavor transition occurring at only the second drop. In each of the two flavor transitions, one side went from grape to tropical punch, while the other did the opposite.

## Flavor preference testing

After the completion of all recording sessions, we conducted two-bottle consumption tests of the Kool-Aid solutions two times over two days for nine of the ten rats. These tests were run in a housing cage different from home-cages and experimental chambers. Tests were 2 min in duration and the location of the bottles was swapped roughly every 20 s to equate time on each side. The flavor and the initial location of the bottles were randomized in rats and swapped between the 1st and 2nd tests.

## Single-unit recording

Wires were screened for activity daily; if no isolable single-unit activity was detected, the rat was removed and the electrode assembly was advanced 40 or 80 μm. Otherwise active wires were selected to be recorded, a session was conducted, and the electrode was advanced at the end of the session. Neural activity was recorded using Plexon Multichannel Acquisition Processor systems (Dallas, TX). Signals from the electrode wires were amplified 20X by an op-amp headstage (Plexon Inc, HST/8o50-G20-GR), located on the electrode array. Immediately outside the training chamber, the signals were passed through a differential pre-amplifier (Plexon Inc, PBX2/16sp-r-G50/16fp-G50), where the single unit signals were amplified 50X and filtered at 150–9000 Hz. The single unit signals were then sent to the Multichannel Acquisition Processor box, where they were further filtered at 250–8000 Hz, digitized at 40 kHz and amplified at 1-32X. Waveforms (>2.5:1 signal-to-noise) were extracted from active channels and recorded to disk by an associated workstation.

## Measures and statistical analyses

Average percent correct and choice latency (defined as the time from the end of odor delivery to withdrawal from the odor port on trials resulting in a correct response) were calculated by trial-type (3-drop, 1-drop, grape, tropical punch) across all trials. The flavor of the reward was defined as that of the first drop.

Units were sorted using Offline Sorter software from Plexon Inc (Dallas, TX). Sorted files were then processed and analyzed in Matlab (Natick, MA). Dopamine neurons were identified via a waveform analysis. Briefly, a cluster analysis was performed based on the half-time of the spike duration and the ratio comparing the amplitude of the first positive and negative waveform segments. The center and variance of each cluster was computed without data from the neuron of interest, and then that neuron was assigned to a cluster if it was within 3 s.d. of the cluster's center. Neurons that met this criterion for more than one cluster were not classified. This process was repeated for each neuron. Neurons were considered putatively dopaminergic if they were in the wide waveform cluster and were also reward-responsive, defined as those that were significant at $p<0.05$ by t-test comparing baseline firing rate with the first 500 ms of reward delivery across all rewarded trials. This waveform analysis is based on criteria similar to that typically used to identity dopamine neurons in

primate studies (*Bromberg-Martin et al., 2010*; *Fiorillo et al., 2008*; *Hollerman and Schultz, 1998*; *Kobayashi and Schultz, 2008*; *Matsumoto and Hikosaka, 2009*; *Mirenowicz and Schultz, 1994*; *Morris et al., 2006*; *Waelti et al., 2001*) and isolates neurons in rat VTA whose firing is sensitive to intravenous infusion of apomorphine or quinpirole (*Jo et al., 2013*; *Roesch et al., 2007*). Neurons identified in this manner are also selectively eliminated by expression of a Casp3 neurotoxin in TH+ neurons in VTA (by infusion of AAV1-Flex-TaCasp3-TEVp into TH-Cre transgenic rats; *Takahashi et al., 2017*).

To calculate difference scores and firing rates for scatter plots, firing rates were aligned to drop delivery and baseline-subtracted using the 500 ms immediately before the light-on at the start of the trial. To capture the peak reward-responsive activity, firing rates from 200 ms to 700 ms after the timestamp for the relevant drop delivery or drop omission were calculated. For number errors, the epochs were aligned to the first omitted drop (at the time the second drop would normally be delivered) in block 2, and the first newly delivered drop (second drop) in block 3. For flavor errors, the epochs were aligned to the first new flavor drop in both blocks 4 and 5. Difference scores were calculated for number transitions as the difference between the average firing rate on the first three rewarded trials in the relevant block and the last five rewarded trials in the same block and direction, and for flavor transitions as the difference between the average firing rate in the first three rewarded trials in the relevant block and the last five trials in the previous block in the same direction.

For the decoding analyses, we used Matlab code from the Neural Decoding Toolbox (www.read-out.info) (*Meyers, 2013*) to construct pseudoensembles consisting of all 30 putative dopamine neurons as described below. Decoding using pseudoensembles has been found to reveal the information held by the activity of populations of neurons in well-learned tasks such as the one we used here as effectively as analyses of real-time simultaneously recorded ensembles (*Rigotti et al., 2013*; *Schoenbaum and Eichenbaum, 1995*). The spike-trains of the 30 neurons were aligned to various trial events (light-on, odor delivery, odor port withdrawal, reward delivery, and light-off), concatenated according to the average time between these events, and then binned into sliding 900 ms bins across the resulting spike-trains. All the correct trials from blocks 4 and 5 were labeled according to the flavor delivered on that trial, with trials from block five labeled according to the flavor of the second drop (the changed drop). The first ten trials in each block for each flavor were then taken from blocks 4 and 5, resulting in 40 total trials for each neuron. This selection resulted in flavor being fully crossed with side (10 trials from each flavor being left-well rewarded and 10 being right-well rewarded). The trials were then randomly divided into 18-20 splits, in each of which there was one test trial of each flavor for each neuron and the remaining 17-19 training trials of each flavor for each neuron. For each split, the flavor of each test trial was classified according to which training set had the highest correlation coefficient with it across the 30 neurons. This random split and test procedure was then repeated 500 times for every epoch to yield the average 1–0 accuracy of the classification at that epoch. This entire procedure was then repeated for sliding sets of 10 trials across the blocks (i.e. trials 1–10 of each flavor in each block, trials 2–11 of each flavor in each block, etc., ending with the last 10 trials of each flavor in each block). The 1–0 accuracy was then plotted separately for test trials taken from block 4 and block 5. The one-tailed 95% confidence interval for chance for the first sliding set of trials was calculated by shuffling the flavor labels 100 times and performing the entire analysis on each resulting dataset.

The decoding analysis shown in *Figure 3c* was similar to that described above, except that only the 900 ms epoch beginning 100 ms after the first new flavor drop was used, test data from blocks 4 and 5 were included together, and the first ten and last ten trials were labeled separately and both included in the same analysis. The resulting classification accuracy was compared with a control classification of flavor in which the identical procedure was followed, except that data from the first drop of block three and the first drop of block five were used. These drops were selected because flavor was unchanged at those drops compared to the previous blocks, because they were part of 3-drop sequences just as in the experimental dataset, and because flavor was crossed with direction just as in the flavor transition analysis. The patterns in the flavor transition vs. flavor unchanged confusion matrices were compared by permutation test in which the flavor labels were shuffled 100 times for each analysis and 100,000 comparisons between the resulting confusion matrices were used to construct a distribution of comparisons. We then calculated the probability that the actual pattern of the two confusion matrices would be observed by chance. That is, we calculated the chance that the differences between flavor transition vs. flavor unchanged in grape early and tropical

punch early would be as great as they were in the real data, while the differences in grape late and tropical punch late would be as small as they were in the real data.

The decoding analysis shown in *Figure 3d* was similar to that described above, except that the decay of decoding accuracy across the block was tested by using a sliding set of trials for both the flavor transition and flavor unchanged analyses. Each curve was then compared to chance by permutation tests with 100 shuffles of the flavor labels each. The accuracy in the unshuffled data was considered significantly greater than chance when it was in the top 5% of the shuffle distribution for five consecutive sliding sets of trials. Average baseline firing rate on the trial-sets included in each of the decoding algorithms was also calculated and shown on *Figure 3d*.

We tested the latency of flavor decoding in the first ten trials of blocks 4 and 5 combined by advancing a 200 ms sliding epoch from the time of new flavor drop delivery until significance (by permutation test, $p < 0.05$) was reached and maintained for at least five consecutive bins. We identified the latency as the end of the first significant epoch.

## Experiment 2

### Subjects

Twenty three human participants (nine male, ages 19–34, mean $\pm$ SD = 25.5$\pm$4.1 years) with no history of psychiatric illness gave informed written consent to participate in this study. The study protocol was approved by the Northwestern University Institutional Review Board.

### Odor stimuli and presentation

Eight food odors, including four sweet (strawberry, caramel, cupcake, gingerbread) and four savory (potato chips, pot roast, sautéed onions, garlic), were provided by International Flavors and Fragrances (New York, NY). For all experimental tasks, odors were delivered directly to participants' noses using a custom-built computer-controlled olfactometer.

### Odor selection and task familiarization

In an initial behavioral testing session, hungry participants (fasted for at least 6 hr) first provided pleasantness ratings of the eight food odors. Based on these ratings, one sweet odor and one savory odor were chosen such that they were matched as closely as possible in pleasantness. Next, we acquired pleasantness ratings for the two selected odors across a range of odor concentrations, diluted to varying degrees with odorless air. Based on these ratings, we selected two concentrations for each odor, such that the two low-concentration odors had the same pleasantness and the two high-concentration odors had the same pleasantness.

Participants next completed 84 trials of the instrumental reversal learning task they would eventually complete in the fMRI scanner. For this task, two abstract visual symbols were randomly chosen to serve as conditioned stimuli (CS) throughout the rest of the experiment. Each trial started with either one of the two CS's (indicating it was a forced choice trial) or a question mark (indicating it was a free choice trial) presented for 4 s. Both CS's were then presented on either side of a center crosshair (side fully randomized and counterbalanced) for 1.5 s, during which time participants were instructed to choose via left or right mouse click the CS that appeared alone in the preceding screen (in the case of a forced choice trial), or whichever CS they preferred (in the case of a free choice trial). If no response was made within 1.5 s, 'TOO SLOW' appeared on the screen and the next trial was initiated after a variable delay. If a response was made, the odor currently paired with the selected CS was delivered after a 2 s delay. Odor delivery, lasting 3 s, was indicated by changing the color of the center crosshair to blue, informing participants to sniff. Participants then rated either the pleasantness or identity of the received odor (rating type randomized), followed by a 0–2 s inter-trial interval.

Across the 84 trials, the choice task was covertly subdivided into 8 blocks of trials delineated by the specific CS-US associations predetermined for that block. Each block consisted of either 9 or 12 trials, and the length of blocks across the session was pseudorandomized. Within a given block, one of the CS's was paired deterministically with the high concentration of one odor identity (e.g., sweet high: $SW_H$), while the other CS was paired deterministically with the low concentration of the same odor identity (e.g., sweet low: $SW_L$). After each block, the CS-US associations were changed without warning, and new blocks always began with two forced choice trials (one for each CS). In the case of

flavor reversals, the flavor of the US was changed for both CS's while leaving CS-value associations the same. In the case of reward value reversals, the CS-value association was swapped between the two CS's, while leaving flavor unchanged. Reversals alternated between flavor and value, and there were seven total reversals across the 84-trial task.

## Choice task during fMRI scanning

The fMRI scanning session was conducted within ~10 days (mean ± SD = 10.0±4.4 days) of the initial behavioral session. During scanning, hungry participants (fasted for at least 6 hr) completed 3 runs of the 84-trial reversal learning task described above. Each run lasted ~21 min, and the sequence of alternating flavor and value reversals was counterbalanced across subjects.

## fMRI data acquisition

MRI data were acquired on a Siemens 3T PRISMA system equipped with a 64-channel head-neck coil. Echo-Planar Imaging (EPI) volumes were acquired with a parallel imaging sequence with the following parameters: repetition time, 2 s; echo time, 22 ms; flip angle, 90°; multi-band acceleration factor, 2; slice thickness, 2 mm; no gap; number of slices, 58; interleaved slice acquisition order; matrix size, 104 × 96 voxels; field of view 208 mm x 192 mm. The functional scanning window was tilted ~30° from axial to minimize susceptibility artifacts in OFC (*Weiskopf et al., 2006*). Each fMRI run consisted of 640 EPI volumes covering all but the dorsal portion of the parietal lobes. To aid in co-registration and normalization of the functional scans, we also acquired 10 EPI volumes for each participant covering the entire brain, with the same parameters as described above except 95 slices and a repetition time of 5.25 s. A 1 mm isotropic T1-weighted structural scan was also acquired for each participant. This image was used for spatial normalization.

## fMRI data preprocessing

All image preprocessing and general linear modeling was done using SPM12 software (www.fil.ion.ucl.ac.uk/spm/). To correct for head motion during scanning, for each subject all functional EPI images across the 3 fMRI runs were aligned to the first acquired image. The motion-corrected images were smoothed with a Gaussian kernel at native scan resolution (2 × 2×2 mm) to reduce noise but retain potential information content (*Gardumi et al., 2016*). For reverse normalization of midbrain regions of interest to participant-specific native space, each participant's T1-scan was normalized to Montreal Neurological Institute (MNI) space using the 6-tissue probability map provided by SPM12. The inverse deformation field resulting from this normalization step was then applied for each participant to a region of interest in MNI space defined by spheres of 4-voxel radius centering on the two midbrain coordinates reported to show a significant univariate response to flavor prediction errors (left: x=-16, y=-14, z=-12; right: x = 6, y=-14, z=-14) (*Howard and Kahnt, 2018*).

## General linear modeling and MVPA analyses

For the decoding analysis, we constructed independent subject-level event-related general linear models (GLMs) for each fMRI run using finite impulse response (FIR) functions specified over 12 time bins time-locked to the onset of each trial. Nuisance regressors included: normalized respiratory activity traces (measured by MR-safe breathing belts affixed around the torso); the six realignment parameters calculated for each scanned image during motion-correction; the derivative, square, and square of the derivative of each realignment regressor; the absolute signal difference between even and odd slices, and the variance across slices, in each functional volume; additional regressors as needed to censor individual volumes in which particularly strong head motion occurred. Odor onsets corresponding to 13 conditions were specified in each GLM: SV→SW reversals, SW→SV reversals, SW and SV 1, 2, 3, and four trials after reversals, SW and SV on the trial immediately preceding reversals, and all other trials. The resulting parameter estimates within a region of interest (ROI) defined by the intersection of an un-normalized anatomical mask of the midbrain and the un-normalized spherical mask described above were extracted for each subject, fMRI run, and condition at the time bin corresponding most closely to odor delivery given hemodynamic lag. Prior to decoding, voxels within each subject's midbrain ROI were sorted according to the difference in responses to flavor transitions on the error trial (combined across SV→SW and SW→SV) and responses on the trial preceding error trials (combined across SW and SV).

The resulting sorted parameter estimates were then submitted to pairwise linear support vector machine decoding analyses using the libsvm implementation (*Chang and Lin, 2011*). Each pairwise analysis corresponded to the SW and SV conditions at a given trial point (i.e., error trial, error trial +1, error trial +2, etc.), and was conducted using a nested cross-validation approach in which we first performed leave-one-subject-out cross-validation in increasing numbers of voxels within the ROI to determine the number of voxels that most effectively decodes reward flavor in a 'training set' of subjects. Leave-one-run-out cross-validated decoding of flavor in the left out subject was then conducted in the number of voxels giving maximal decoding accuracy from the training set of subjects. This process was repeated for each subject, resulting in an independent decoding accuracy value calculated for each subject and decoding pair.

An identical analysis was conducted for value transitions (i.e., flavor unchanged), in which GLM's were specified using the same condition types time locked to these type of reversals: SW and SV at the value error trial, SW and SV at 1, 2, 3, and four trials after value reversal and immediately before value reversal, and all other trials. We then implemented the same nested cross-validation method to generate decoding accuracies for pairwise tests at each trial point.

The patterns in the flavor transition vs. flavor unchanged confusion matrices were compared by permutation test. For each analysis and subject we shuffled the condition labels (first sweet, first savory, last sweet, last savory, within each run) 100 times. For each of these shuffled permutations, we conducted the decoding analysis to generate a confusion matrix. We then randomly sampled one of these confusion matrices for each subject 100 times and averaged the sampled matrices across subjects to generate 100 population averages. We then randomly sampled from these 100 averages to generate 100,000 comparisons between the two matrices to generate a distribution of comparisons. From this distribution, we calculated the probability that differences between flavor transition vs. flavor unchanged in sweet error trial and savory error trial were as great as they were in the real data, and differences in sweet last trial and savory last trial were as small as they were in the real data.

## Acknowledgements

This work was supported by the Intramural Research Program at the National Institute on Drug Abuse and National Institute on Deafness and Other Communication Disorders grant R01DC015426 (to TK). The opinions expressed in this article are the authors' own and do not reflect the view of the NIH/DHHS.

## Additional information

### Competing interests

Geoffrey Schoenbaum: Reviewing editor, *eLife*. Thorsten Kahnt: Reviewing editor, *eLife*. The other authors declare that no competing interests exist.

### Funding

| Funder | Grant reference number | Author |
|---|---|---|
| National Institute on Drug Abuse | ZIA-DA000587 | Geoffrey Schoenbaum |
| National Institute on Deafness and Other Communication Disorders | R01DC015426 | Thorsten Kahnt |

The funders had no role in study design, data collection and interpretation, or the decision to submit the work for publication.

### Author contributions

Thomas A Stalnaker, Conceptualization, Data curation, Formal analysis, Supervision, Investigation, Visualization, Methodology, Writing—original draft, Writing—review and editing, Experiment 1 in

rats: designed the experiment with YKT and GS and analyzed the data, Experiment 2 in humans: provided input on approaches and interpretation with SJG and GS, Wrote the manuscript with JDH, TK and GS with input from all of the other authors; James D Howard, Conceptualization, Data curation, Formal analysis, Investigation, Visualization, Methodology, Writing—review and editing, Experiment 2 in humans: designed, conducted, and analyzed the experiment with TK, Wrote the manuscript with TAS, TK and GS with input from all of the other authors; Yuji K Takahashi, Conceptualization, Data curation, Formal analysis, Investigation, Visualization, Methodology, Writing—original draft, Writing—review and editing, Experiment 1 in rats: designed the experiment with TAS and GS, Conducted the experiment; Samuel J Gershman, Conceptualization, Writing—original draft, Writing—review and editing, Experiment 1 in rats: provided input on approaches and interpretation with TK and GS, Experiment 2 in humans: provided input on approaches and interpretation with TAS and GS; Thorsten Kahnt, Conceptualization, Resources, Data curation, Formal analysis, Supervision, Funding acquisition, Investigation, Visualization, Methodology, Writing—original draft, Writing—review and editing, Experiment 1 in rats: provided input on approaches and interpretation with SJG and GS, Experiment 2 in humans: designed, conducted and analyzed the experiment with JDH, Wrote the manuscript with TAS, JDH and GS with input from all of the other authors; Geoffrey Schoenbaum, Conceptualization, Resources, Supervision, Funding acquisition, Investigation, Methodology, Writing—original draft, Project administration, Writing—review and editing, Experiment 1 in rats: designed the experiment with YKT and TAS, provided input on approaches and interpretation with TK and SJG, Experiment 2 in humans: provided input on approaches and interpretation with TAS and SJG, Wrote the manuscript with TAS, JDH and TK with input from all of the other authors

### Author ORCIDs
Thomas A Stalnaker (iD) https://orcid.org/0000-0003-4402-5448
James D Howard (iD) https://orcid.org/0000-0002-9309-3773
Samuel J Gershman (iD) https://orcid.org/0000-0002-6546-3298
Thorsten Kahnt (iD) https://orcid.org/0000-0002-3575-2670
Geoffrey Schoenbaum (iD) https://orcid.org/0000-0001-8180-0701

### Ethics
Human subjects: Subjects gave informed consent to participate in the experiment. The protocol (#STU00098371) and consent forms were approved by Northwestern University's Institutional Review Board.
Animal experimentation: This study was performed in strict accordance with the recommendations in the Guide for the Care and Use of Laboratory Animals published by The National Research Council of The National Academies. All of the animals were handled according to approved animal care and use committee protocols of the NIH. The protocol (#15-CNRB-108) was approved by the NIDA-IRP ACUC. All surgery was performed under isoflurane anesthesia, and every effort was made to minimize suffering.

### Decision letter and Author response
Decision letter https://doi.org/10.7554/eLife.49315.009
Author response https://doi.org/10.7554/eLife.49315.010

## Additional files

### Supplementary files
• Transparent reporting form  DOI: https://doi.org/10.7554/eLife.49315.007

### Data availability
The raw data that went into the analyses shown in Figures 1, 2 and 3 are archived at https://github.com/tastalnaker/dopamine_ensemble_analysis.git (copy archived at https://github.com/elifesciences-publications/dopamine_ensemble_analysis).

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
