## [Decision Letter]

**Acceptance summary:**

In this study, the authors demonstrate that dopamine neuron ensembles encode the specific identity of surprising outcomes. This work is an extension, and more in depth analysis, of two previously published studies by these research groups who previously demonstrated that (putative) dopamine neurons or midbrain BOLD signals do not only signal surprising changes in value (as generally assumed), but also signal sensory prediction errors (i.e. value-neutral violations in expected outcome identity). Here, the authors take this a step further, using a decoding analysis to show that the firing pattern in putative dopamine ensembles or midbrain fMRI signals also encode the specific identity of the surprising outcome. This has potentially far-reaching implications for our understanding of the role of dopamine in learning.

**Decision letter after peer review:**

Thank you for submitting your article "Dopamine neuron ensembles signal the content of sensory prediction errors" for consideration by *eLife*. Your article has been reviewed by three peer reviewers and the evaluation has been overseen by Kate Wassum as the Senior Editor. The reviewers have opted to remain anonymous.

The reviewers have discussed the reviews with one another and the Reviewing Editor has drafted this decision to help you prepare a revised submission.

Essential revisions:

You will see that the reviewers were very enthusiastic about the paper, but raised a number of overlapping concerns. We expect you will make good faith effort to address each of the concerns noted in the appended reviews. We especially direct your attention to the following:

1) Discuss the problem of dimensionality (i.e., the vastness of potential sensory prediction errors relative to the number of dopamine neurons; reviewer 2 point 2) and of the limitations of using 2 outcomes in this task.

2) Discuss of the limitations of the decoding analysis approach (reviewer 3 point 1) including whether or how this information might be read out in the brain (reviewer 2 point 1).

3) Apply the decoding analysis to the value prediction error phase of the task to ask whether DA/midbrain ensembles encode outcome identity when the presence of the outcome altogether, not only its identity, is surprising (reviewer 1 point 1).

4) Discuss how these data are compatible (or not) with evidence that optogenetic activation, which activates DA neurons, but disrupts the precise firing pattern of DA ensembles, can promote learning about the sensory features of an event (reviewer 1 point 3).

Reviewer #1:

In this study, Stalnaker et al. make the original and thought-provoking argument that dopamine neuron ensembles encode the specific identity of surprising outcomes. This work is an extension, and more in depth analysis, of two previously published studies by these two research groups (Schoenbaum and Kahnt) who independently demonstrated that (putative) dopamine neurons do not only signal surprising changes in value (as generally assumed), but also signal sensory prediction errors (i.e. value-neutral violations in expected outcome identity). Here, the authors take it a step further. Using a decoding analysis, they present converging evidence from single cells recording in rats (experiment 1) and fMRI in humans (experiment 2) showing that the firing pattern in dopamine ensembles also encode the specific identity of the surprising outcome. This has potentially far-reaching implications for our understanding of the role of dopamine in learning.

The results are compelling, data are presented in a manner that is easy to understand, interpretations are logical, and the manuscript is very well written. The inclusion of rodent and human work in the same paper is commendable. Although some results from these datasets have already been published, the type of analyses and the conclusions derived from these analyses are clearly novel, and well beyond what was previously published, and justifies a separate publication.

- Encoding of sensory information in "classic" reward prediction errors?

It seems that the decoding analysis was applied strictly to the portion of the task that features identity prediction errors (value-neutral switch from O1 to O2). But what about the portion of the task that features "classic" reward prediction errors? Is there also evidence that DA ensembles encode outcome identity when the presence of the outcome altogether, not only its identity, is surprising (arguably the most common scenario when organisms learn about outcomes)?

- Functional role for sensory information in DA prediction errors?

Authors propose that the information about the identity of a surprising outcome contained in the firing pattern of DA ensembles, instructs downstream areas what to learn. The authors should highlight how this hypothesis is a significant shift from the way prediction errors (even sensory PE) are generally conceived (i.e. as permissive signals for learning, with the content of learning being determined by activity in downstream regions, e.g. Glimcher, 2011). The authors should also consider discussing how this new hypothesis fits with the highly divergent and overlapping nature of DA projections (Arbuthnott and Wickens, 2007; Matsunda et al., 2009; Bolam and Pissadaki, 2012).

- The authors cite studies showing that DA firing promote learning about sensory features of an event. Some of these studies (Sharpe et al., 2017; Keiflin et al., 2019) used optogenetic activation, which activates DA neurons as a whole but most likely disrupts the precise firing pattern of DA ensembles. How is that compatible with the idea the precise pattern of firing in DA ensembles instruct the content of learning?

Reviewer #2:

This paper studies the population activity of dopamine neurons in rodents and midbrain BOLD in humans and shows that it is possible to decode more sophisticated aspects of predictive failures than would be implied just by changes in scalar value. In particular, a change from an outcome of one flavor (rodents) or sweet vs. savory (humans) is decodeable from this population activity.

In short, this is a very well conducted pair of studies, and the results and the analyses are convincing. The paper relates well to previous studies looking at less scalar aspects of the dopamine signal, adding an interesting new perspective.

My main concern is that the underlying implications are not so clear:

- Many dopamine cells have exuberant axonal arbors (although for sure, more in the SNc than the VTA) – thus the experimenter's ability to decode this signal from individual cells may vastly outweigh the ability of downstream neurons or tissue to do this decoding. It would be good to be convinced that this signal had a way of actually mattering. The fMRI results don't constitute such a proof.

- Related to this – what my computational colleagues refer to as the dimensionality of the problem seems unfavorable. The number of possible juices, let alone things of equivalent value, that could be substituted, is vast. They have elaborated representations in various regions of the cortex which is sized to match. However, there aren't many dopamine neurons. Thus, the fact that there is a reliably decodable signal – coming, for instance, from the upstream sampling properties of individual dopamine neurons (e.g., if they get somewhat random collections of inputs) – when there are only two choices, and the chance to build a sophisticated decoder bears little on the real problem of making state- rather than value-based prediction errors be useful.

Reviewer #3:

This study examines the role of dopamine neurons in sensory prediction errors, which are signals that indicate an event that violates expectancies, independent of the value of the event. The authors show that the response of single dopamine neurons detects sensory violations, but they don't necessarily discriminate between different kinds of violations. However, at the population level it is possible to decode the nature of the violation. In addition, they show that a similar pattern of results can be obtained in humans using fMRI. The study is a solid contribution to the literature and its interpretation is relatively straightforward. The task is well-designed, the analyses are appropriate, and the results are clear.

1) The Discussion should do more to couch the interpretation of the results. A danger with decoding analyses is that just because something can be decoded, it doesn't necessarily mean that the brain is using that information. For example, it would be relatively straightforward to decode line orientation from retinal ganglion cells, even though we know that these cells are not encoding this information.

2) Figure 3A: The mountain plot obscures the data. I'd suggest just plotting the top-down view and allowing the color to represent decoding accuracy.

3) Figure 3C: I don't know why this data has been spatially smoothed. Confusion matrices are not continuous.

4) Figure 4D: The description in the figure legend is confusing. The first sentence states that decoding accuracy was above chance on the error trial, but then the next sentence states that it was not above chance on this trial.

5) Figure 4E: Unlike Figure 3C there are no stats to support the claims as to what data is or is not significant.

---

## [Author Response]

Essential revisions:You will see that the reviewers were very enthusiastic about the paper, but raised a number of overlapping concerns. We expect you will make good faith effort to address each of the concerns noted in the appended reviews. We especially direct your attention to the following:1) Discuss the problem of dimensionality (i.e., the vastness of potential sensory prediction errors relative to the number of dopamine neurons; reviewer 2 point 2) and of the limitations of using 2 outcomes in this task.

We thank the reviewers for all their work in reading our original paper, their kind comments of support, and their excellent and insightful questions. We have tried to address them as best we can with the existing data and within the constraints of the manuscript, but we hope they will appreciate that some of the questions raised are almost existential and thus hard to arrive at a fully satisfying text response. We have done our best.

With regard to the problem of dimensionality (reviewer 2, point 2), we definitely appreciate this issue. However, it is undeniable that our brains do deal with this problem. And thus the solution must be in our heads. While we do not argue that our small experiment provides the full solution, it is a step in that direction to show that this system, which is already accepted as serving as a teaching signal in one critical dimension (value), appears to contain information to do so in another completely orthogonal dimension (flavor). The odds are low that we stumbled upon the only other dimension in the world that activates this system, and so if we accept that many other dimensions may be represented similarly, then the current results suggest the dopamine system may broadly provide information about unexpected events in the world. And, critically, we think the system has the capacity to do this. There are approximately 40,000 dopamine neurons in the VTA of rats, and another 25,000 in SN (Nair-Roberts, 2008). In humans, the total number is about 300,000 (Hirsch et al., 1988). If we assume that each neuron only provides a single bit of information, this means the capacity of just the VTA in rats is 2^40,000. Of course, this calculation is obviously problematic as it ignores certain redundancy in any coding, yet even if we reduce the cell number to 1000 real bits of information, we still end up with 1.0715e+301 potential patterns. This would cover an unimaginably large number of possible sensory features and events. And of course, information represented in spiking may be augmented (or attenuated) by factors such as co-release of other neurotransmitters, location (region, cell type, dendritic compartments) and type (receptors, second messenger cascades) of interactions with post-synaptic neurons, etc. Even if all this combines to yield only 20 or 30 uniquely coded dimensions, we still end up with a billion possible patterns of output. And this enormous number assumes a binary code; it seems likely to us that a single neuron does more than signal yes/no. This number seems big enough, with assistance from other systems (we are not arguing this is the only learning signal) and with contextual modulation of the signal (i.e. some dimensions of the environment might be given priority or not, depending on the situation, by modulating inputs), to deal with much of the problem of dimensionality.

The potential importance of contextual modulation of the processing to reduce dimensionality is worth emphasizing. The hippocampus is perhaps the most famous example a system or brain area where we understand its function most clearly by considering ensemble-based coding. This area is thought not only to represent location in space, but to integrate this information with other dimensions of information (head direction, direction of movement, reward, internal motivation, and so on). Further, the same structure is now generally accepted as representing non-spatial information similarly. Yet it is not thought that it holds all this information simultaneously online about every place in the world at once, let alone every continuous variable. Rather, it is thought to do so in a way that is constrained by inputs regarding the current context and goals of the subject. Here, similarly, the dopamine system is presumably governed by the content of the inputs from upstream areas directing its processing to certain variables or dimensions of information. Of course, here the problem is different as the system is tasked with registering unexpected events. But one could imagine a major reduction in dimensionality if upstream areas ignored many aspects of the environment deemed to be stable and irrelevant unless they passed some threshold, particularly in highly structured settings such as experimental tasks where so much of the world can be categorized as irrelevant. For instance, imagine all the things in the training chamber that the rat has learned to ignore and that presumably would not evoke any changes in dopamine firing unless there was a sudden large deviation in them (temperature, ambient lighting, external sounds of the experimenter, smells here and there, and so on). Indeed we have some evidence of this – in our original report (Takahashi et al., 2017), we found that the dopamine neurons only exhibited sensory prediction errors when there was some change in the rats’ behavior when the reward identity switched. To be clear, this behavioral change was not directional and thus not value-based. But it indicated objectively that the rat had noticed the change in flavor. When there were no changes in behavior that we could detect at the transition, the same neurons did not show error signals. This is potential evidence of a shutting off or modulation of the dimensions responded to by the dopamine neurons based on the rats’ subjective goals or perceptions of the environment. Such a mechanism could drastically simplify the dimensionality problem.

We have edited the Discussion to include a paragraph that makes some of the above points in a limited fashion.

Regarding whether the content of the sensory prediction error upon a switch from A->B reflects the presence of B or the absence of A, we are not equipped to test this with data from the experiments presented here because they only involved two distinct reward identities. However, we agree that this is an excellent and very important question, and we are currently planning experiments to get directly at this question. For now, we have included a paragraph in the Discussion noting the issue.

Meanwhile, to try to satisfy this concern a bit, we have looked at data from another study on sensory prediction errors in humans (Suarez et al., 2019) in which we used four distinct value-matched food odors as rewards in a similar reversal learning task (A, B, C, D). This allowed us to conduct a reanalysis of this dataset in which we attempted to decode the identity of the prediction error on trials in which only one of these types of information was in error. For instance, we examined activity on A->B versus A->C switches to determine if the identity of the unexpected outcome delivered could be decoded, and we examined activity on A->B versus C->B switches to test if the identity of the omitted outcome could be decoded. We found above-chance decoding accuracy in the former scenario (t_18_=2.19, *p*=0.042), but chance accuracy in the latter (t_18_=0.89, *p*=0.39).

These results indicate that the sensory prediction error signal encoded in ensemble midbrain activity, at least in humans, reflects the content of the *present* unexpected outcome. However, we are reluctant to conclude that the omitted outcome is not also represented. If the absence of expected sensory information is represented by reductions in patterns of activity, similar to the absence of expected value, it may be much harder or require greater statistical power to see, especially against the backdrop of delivered actual events. That is, when something new is delivered, it is not clear that the omission of the thing replaced registers as strongly, thus the error pattern may be biased to reflect the actual unexpected event that occurs. Further, Suarez et al. involved one long run of fMRI data collection rather than distinct runs that would be most appropriate for constructing separate independent training and test sets for decoding. We therefore feel that these results are somewhat tenuous and would prefer to not include them in the present manuscript. We hope the paragraph in the Discussion alluding to this outstanding question is sufficient for now.

2) Discuss of the limitations of the decoding analysis approach (reviewer 3 point 1) including whether or how this information might be read out in the brain (reviewer 2 point 1).

Reviewer 3 makes the good point that we cannot be certain from our decoding results that the regions downstream of dopamine neurons are actually using the outcome identity information that we can decode from their activity. This is a hazard of any correlative study, it seems to us, not just a problem for decoding analyses. However, in our results, unlike the retinal cell example cited by the reviewer, the relevant information can be decoded only when the hypothesis predicts it would be useful to downstream regions (i.e. when it is unexpected), and not at other times. If one could decode line orientation from retinal cell activity *only* when line orientation was useful to the rest of the brain, that would be much stronger evidence of functional relevance at the level of the retina. Although we cannot be sure outcome identity information held by dopamine ensembles is functionally relevant for error-based learning, the data are at least consistent with this proposal. Further, there is in fact evidence that manipulating patterns of dopamine activity causes sensory error-based learning (also see the answer to essential revision 4 below for a related discussion of how such a manipulation may produce this learning). These studies include Sharpe et al., 2017, and perhaps even more obviously the study by Keiflin et al. (2019), in which stimulation of dopamine neurons at the time of reward in a blocking procedure unblocks devaluation-sensitive reward learning (indicating dopamine stimulation drove the development of an association between the CS and the sensory properties of the US, such as its flavor). Admittedly, the pattern of activity created by optogenetic manipulation in these studies was not specifically known, but they clearly show that a change in the pattern can be functionally relevant for learning. We have added a paragraph to the Discussion to clearly acknowledge this limitation and point out the relationship to the other studies that have found a functional relevance of dopamine release to sensory associative learning.

Regarding the question of how information held by dopamine ensemble firing rates could be read out by downstream regions, we must acknowledge that our study was not designed to address this question (as would be true of any neural recording study). Reviewer 2 mentions the important issue of exuberant axonal arborization of dopamine neurons. This is perhaps related to the broader idea, commonly held for many years, that dopamine operates by volume transmission, which would not maintain the fine structure of ensemble firing rates. However, the exclusivity of dopaminergic volume transmission is being challenged in recent literature (e.g. Liu and Kaeser, Current Opinion in Neurobiology, 2019). And regarding the specific issue of axonal arborization, whether it makes a readout of firing rate information difficult depends on the specific pattern of arborization and release points coming from each dopamine neuron. We would argue that the question of how much and what information is carried by dopamine release has yet to be well-tested, in large part because the techniques for tracking release have lacked the spatial resolution to ask this question (until recently). In future studies, we intend to address this question using recently developed imaging techniques, such as using the DLight fluorescent dopamine sensor with an in vivo miniscope.

3) Apply the decoding analysis to the value prediction error phase of the task to ask whether DA/midbrain ensembles encode outcome identity when the presence of the outcome altogether, not only its identity, is surprising (reviewer 1 point 1).

In response to the first point, we would observe that, given the task design, it is unclear whether the flavor (or identity) is unexpected across transitions designed to elicit “classical” reward prediction errors, in both the rat and human version of the task. Further, it is reasonable (and c/w behavioral data) to think that the “value” represented by the additional drops (in the rat task) or increased intensity of a food odorant (in the human case) might swamp any attention to what precise flavor those drops have or what specific qualities the food odorant has. Consistent with this idea, we do not find evidence that identity per se is decoded across these transitions. In the human case, that data was already shown in the original manuscript in Figure 4D-E (admittedly this might not have been clear, because we labeled this transition “flavor unchanged,” but in fact it tested flavor decoding across value transitions – we have now clarified this in the figure).

In the rat case, the data in the “flavor unchanged” control condition did not test this idea, because it included only the first drop of milk, which would have been fully expected across the value transitions. However, in response to this concern, we did run an additional analysis, testing flavor decoding on the second drop of milk (i.e. the new one) across these value transitions. This analysis, presented in Author response image 1, did show somewhat elevated decoding of flavor early in the block. However, in line with the human data, this decoding was not significantly above chance. We have chosen not to include these data in the revised manuscript, because (1) the above reasons suggest it is not the best test of this question, and (2) the residual decoding, though not significantly greater than chance, might result from the ensemble decoding the location of the milk delivery, rather than the flavor. In our design, this analysis confounds “side” with “flavor”; whereas in both the value transitions and the flavor transitions, side is counterbalanced with the tested factors. As discussed below, we have evidence that the location of the reward is another sensory feature possibly signaled by dopamine neurons.

Regarding the “selectivity of information encoded” comment, we conducted additional analyses to test whether we could decode the identity of the predictive cue or the response made (side in the rat task, L/R button press in the human task) from midbrain patterns of activity at the time these events occurred. In the rat data, decoding accuracy was significantly greater than chance for cue identity across the entire block (see Author response image 2) and for “side” across flavor transitions (see Author response image 2). It could be argued that cue identity is always surprising to the rats, because trial-types are pseudorandomly chosen, thus the presence of this signal undiminished across the block potentially makes sense. And side or location is arguably a feature of the reward, and thus it is reasonable that the error code would include information not only about the flavor of the new reward but also that the flavor is appearing in a new location.

However, in the human data, decoding accuracy was at chance for both analyses (cue identity: t_22_=0.24, p=0.81; side: t_22_=0.77, p=0.45). Whether that reflects a problem with the results or a difference in species, task (e.g., side of port vs. left or right button press), or the detail available in BOLD response versus single units, we do not know. And obviously the experiments were not rigorously designed to address the encoding of errors in other information in any case, the way they were designed to test encoding of flavor errors. For these reasons, we have not included these analyses in the revision, since they could be interpreted and potentially misinterpreted in many ways. We would prefer to leave these questions for future, more properly designed studies.

**Author response image 2. respfig2:** 

4) Discuss how these data are compatible (or not) with evidence that optogenetic activation, which activates DA neurons, but disrupts the precise firing pattern of DA ensembles, can promote learning about the sensory features of an event (reviewer 1 point 3).

We agree this is an excellent point. And in our opinion it is a question that is at issue in nearly every optogenetic study of which we are aware (but see Jennings et al., 2019). Specifically, how can the random activation (and in our opinion even inhibition) of neurons engaged in representing information through a meaningful pattern of activity (as we generally accept the entire brain is doing) cause anything meaningful and expected to happen? This, to us, is a fundamental and nearly entirely ignored issue at the heart of the field’s current hyperventilation over optogenetics. We do not pretend to know the answer to this conundrum. However, in our particular case, we would offer the following speculation. Although in isolation the pattern of activation in studies such as Sharpe et al. (2017) would surely be random, owing to factors governing viral expression and light penetration, we believe that in the controlled settings we have used, it can actually recapitulate a somewhat normal pattern of firing due to the presence of well-controlled external events during the stimulation. That is, even though the outcomes are somewhat expected in the blocking design used in Sharpe et al., they are still there. And as a result, the input reflecting their presence is still impinging on the system, although the effect of this input cannot be detected in the output of the dopamine neurons. One might imagine that there could be a subthreshold or “ghost” pattern of activity in dopamine neurons that reflects the presence of this outcome. By randomly injecting current across a subset of this population, our stimulation could recover from this “ghost" pattern one sufficiently similar to the one that would have occurred had the same event been unexpected. As a result, it would reinstate apparently normal learning, given the very simple behavioral readout we use. We are now running studies to look at the neural similarity of such an artificial memory to what is normally learned, but that is our speculation for how to reconcile the current data with those prior results. We have now added this idea to the Discussion.

Reviewer #1:[…]- Encoding of sensory information in "classic" reward prediction errors?It seems that the decoding analysis was applied strictly to the portion of the task that features identity prediction errors (value-neutral switch from O1 to O2). But what about the portion of the task that features "classic" reward prediction errors? Is there also evidence that DA ensembles encode outcome identity when the presence of the outcome altogether, not only its identity, is surprising (arguably the most common scenario when organisms learn about outcomes)?

Please see the response to essential revision 3 above.

- Functional role for sensory information in DA prediction errors?Authors propose that the information about the identity of a surprising outcome contained in the firing pattern of DA ensembles, instructs downstream areas what to learn. The authors should highlight how this hypothesis is a significant shift from the way prediction errors (even sensory PE) are generally conceived (i.e. as permissive signals for learning, with the content of learning being determined by activity in downstream regions, e.g. Glimcher, 2011). The authors should also consider discussing how this new hypothesis fits with the highly divergent and overlapping nature of DA projections (Arbuthnott and Wickens, 2007; Matsunda et al., 2009; Bolam and Pissadaki, 2012).

We have added to our discussion of the novelty of this idea in our Introduction, referencing Glimcher, 2011 as well as other papers in discussing how this signal would differ from salience. We apologize but we were not sure how to incorporate the other citations. We agree that the implementation or effectiveness of any teaching signal in the spiking would be affected by the distribution and sparsity of the connections to the downstream neurons, and we have generally noted the importance of this in responding to essential revision 2 above. Doing more, especially without knowing specifically what the reviewer is suggesting, seems beyond the scope of the study.

- The authors cite studies showing that DA firing promote learning about sensory features of an event. Some of these studies (Sharpe et al., 2017; Keiflin et al., 2019) used optogenetic activation, which activates DA neurons as a whole but most likely disrupts the precise firing pattern of DA ensembles. How is that compatible with the idea the precise pattern of firing in DA ensembles instruct the content of learning?

Please see the response to essential revision 4 above.

Reviewer #2:[…]My main concern is that the underlying implications are not so clear:- Many dopamine cells have exuberant axonal arbors (although for sure, more in the SNc than the VTA) – thus the experimenter's ability to decode this signal from individual cells may vastly outweigh the ability of downstream neurons or tissue to do this decoding. It would be good to be convinced that this signal had a way of actually mattering. The fMRI results don't constitute such a proof.

Please see the response to essential revision 2 above.

- Related to this – what my computational colleagues refer to as the dimensionality of the problem seems unfavorable. The number of possible juices, let alone things of equivalent value, that could be substituted, is vast. They have elaborated representations in various regions of the cortex which is sized to match. However, there aren't many dopamine neurons. Thus, the fact that there is a reliably decodable signal – coming, for instance, from the upstream sampling properties of individual dopamine neurons (e.g., if they get somewhat random collections of inputs) – when there are only two choices, and the chance to build a sophisticated decoder bears little on the real problem of making state- rather than value-based prediction errors be useful.

Please see the response to essential revision 1 above.

Reviewer #3:[…]1) The Discussion should do more to couch the interpretation of the results. A danger with decoding analyses is that just because something can be decoded, it doesn't necessarily mean that the brain is using that information. For example, it would be relatively straightforward to decode line orientation from retinal ganglion cells, even though we know that these cells are not encoding this information.

Please see the response to essential revision 2 above.

2) Figure 3A: The mountain plot obscures the data. I'd suggest just plotting the top-down view and allowing the color to represent decoding accuracy.

To address this comment and the similar one made by reviewer 2, we have rotated the surface plots so that the obscured parts of the plots are visible. In our view, the rotated surface plots illustrate the data well. However, we would happily substitute flat heat plots if the reviewers still believe the surface plots are inadequate.

3) Figure 3C: I don't know why this data has been spatially smoothed. Confusion matrices are not continuous.

We apologize for this problem. Although the data in the confusion matrices were not spatially smoothed, there was some sort of graphics distortion affecting those panels which was apparent only on some computers (which is why we didn’t notice it before submission). We believe we have corrected it in the revised figures; however, if you still see any apparent smoothing on your device, please let us know and we will take further steps to correct the problem.

4) Figure 4D: The description in the figure legend is confusing. The first sentence states that decoding accuracy was above chance on the error trial, but then the next sentence states that it was not above chance on this trial.

We apologize for the lack of clarity. The first sentence referred to error trials on flavor transitions (black line), whereas the second sentence referred to value transitions (gray line). We have now clarified the figure legend.

5) Figure 4E: Unlike Figure 3C, there are no stats to support the claims as to what data is or is not significant.

We have added the permutation test to of the legend for Figure 4E and describe this test in the Materials and methods section. We also indicate in Figure 4D which data are significantly different from chance.